# Antiplatelet Therapy in Atrial Fibrillation Patients on Direct Oral Anticoagulants Undergoing Percutaneous Coronary Intervention: Which and How

**DOI:** 10.3390/jcm14176331

**Published:** 2025-09-08

**Authors:** Luca Martini, Matteo Lisi, Graziella Pompei, Manfredi Arioti, Francesco Bendandi, Michael Y. Henein, Matteo Cameli, Andrea Rubboli

**Affiliations:** 1Department of Medical Biotechnologies, Division of Cardiology, University of Siena, 53100 Siena, Italy or m.henein@imperial.ac.uk (M.Y.H.); cameli@unisi.it (M.C.); 2Department of Emergency, Internal Medicine and Cardiology—AUSL Romagna, Division of Cardiology, S. Maria delle Croci Hospital, 48121 Ravenna, Italy; matteo.lisi@auslromagna.it (M.L.); graziella.pompei@auslriomagna.it (G.P.); manfredi.arioti@auslromagna.it (M.A.); francesco.bendandi@auslromagna.it (F.B.); 3Cardiovascular Institute, Azienda Ospedaliero—Universitaria of Ferrara, 44121 Ferrara, Italy; 4Imperial College London, London SW3 6LY, UK; 5Faculty of Medicine, University of Bologna—Campus of Ravenna, 48121 Ravenna, Italy

**Keywords:** atrial fibrillation, oral anticoagulant therapy, anti-thrombotic therapy

## Abstract

In patients on oral anticoagulation (OAC), typically for atrial fibrillation (AF), undergoing percutaneous coronary intervention (PCI), the antiplatelet drugs to be added to direct oral anticoagulant (DOAC) are aspirin and clopidogrel during the initial, short (up to one week) period of triple antithrombotic therapy (TAT), and clopidogrel alone during the subsequent 6- to 12-month period of double antithrombotic therapy (DAT). Both direct and indirect data support the recommendation to avoid the more potent P2Y12 inhibitors—ticagrelor and prasugrel—as part of TAT, owing to the increased risk of bleeding. There is less and inconclusive data available regarding the safety and efficacy of DAT when ticagrelor or prasugrel are used instead of clopidogrel. Also, there is very limited evidence for the use of aspirin instead of clopidogrel in a DAT regimen. While acknowledging the safety and effectiveness of the recommended strategies above, it would, nonetheless, be valuable to have alternative options in the choice of antiplatelet agents. In case of very high thrombotic risk, especially when stents are positioned in potentially risky sites (such as the left main or last remaining vessel) a more potent P2Y12 inhibitor than clopidogrel may be warranted. Moreover, non-responsiveness to, or pharmacological interactions of, clopidogrel may hamper its efficacy. In this review, we aim at presenting and discussing the evidence supporting the current recommendations for the use of the various antiplatelet agents in AF patients on OAC undergoing PCI, as well as at giving a glimpse at future perspectives.

## 1. Introduction

For patients on oral anticoagulation (OAC), typically for atrial fibrillation (AF), who require percutaneous coronary intervention (PCI), current antithrombotic guidelines recommend an initial short course (up to one week) of triple antithrombotic therapy (TAT) consisting of a direct oral anticoagulant (DOAC)—such as apixaban, dabigatran, edoxaban, or rivaroxaban—and dual antiplatelet therapy (DAPT), followed by a period of 6 to 12 months of double therapy (DAT) with a DOAC and single antiplatelet therapy (SAPT), and then indefinite DOAC monotherapy [1,2,3]. The DAPT regimen to be administered in the context of TAT should include aspirin and clopidogrel, but not prasugrel or ticagrelor, as the P2Y12 inhibitor [1,2,3]. During subsequent DAT, the single antiplatelet agent to be combined with DOAC should, preferably, be clopidogrel [1,2,3]. This review aims to present and discuss the evidence supporting current recommendations on the use of various antiplatelet agents in AF patients on OAC who undergo PCI, and to explore future perspectives.

## 2. Current Evidence for the Combination of P2Y12 Inhibitors and Direct Oral Anticoagulants

In the four randomized clinical trials, PIONEER AF-PCI [4], RE-DUAL PCI [5], AUGUSTUS [6], and ENTRUST AF-PCI [7], comparing TAT and DAT strategies in patients with AF undergoing PCI and/or who are hospitalized for acute coronary syndrome (ACS), a total of 10,969 patients were enrolled (Table 1). Patients were randomized to either TAT with warfarin, aspirin, and P2Y12 inhibitor or DAT with DOAC and P2Y12 inhibitor (4–7) (Table 1). Across the four studies, clopidogrel accounted for nearly all the P2Y12 inhibitor use in combination with OAC and aspirin in the TAT arm or OAC alone in the DAT arm [4,5,6,7] (Figure 1). The proportion of patients receiving prasugrel or ticagrelor instead of clopidogrel was in the order of 1% and 4–12%, respectively, with an overall number of 892 patients [4,5,6,7] (Figure 1). This strong evidence supports the current guideline’s recommendation for the use of clopidogrel as the P2Y12 inhibitor of choice in combination therapies, either TAT or DAT [1,2,3]. Additional considerations arise from the results of the TRITON-TIMI 38 [8] and PLATO [9] trials where patients with ACS (undergoing PCI in the majority of cases) were randomized to DAPT with aspirin and prasugrel or ticagrelor, respectively, versus conventional DAPT with aspirin and clopidogrel. The significant increase in non-coronary artery bypass graft-related bleeding observed with prasugrel and ticagrelor when compared with clopidogrel in the TRITON-TIMI 38 [8] and PLATO [9] studies, respectively, is a further indirect deterrent for the use of the more potent P2Y12 inhibitors in patients who are inherently at high bleeding risk because of OAC [10]. Further data in support of the preferential use of clopidogrel in combination therapies are derived from the observational and prospective TRANSLATE-ACS registry [11], which was carried out in 233 hospitals in the United States and in which 11,756 patients with acute myocardial infarction treated with primary PCI were enrolled. In this population, the 6-month risk of any BARC-defined bleeding was significantly higher among patients discharged on TAT of OAC, predominantly with warfarin but also including dabigatran and rivaroxaban, plus DAPT (617 patients, 5.3%) compared to the remaining patients who were on DAPT alone [11]. Moreover, within the TAT group, patients receiving prasugrel (91 patients, 0.8%) instead of clopidogrel (526 patients, 4.5%) experienced a significantly greater incidence of bleeding (Adjusted Incidence Rate Ratio [RR] 2.37; 95% Confidence Intervals [CI] 1.36–4.15; *p* = 0.003) [11]. A meta-analysis of the four randomized clinical trials, PIONEER AF-PCI [4], RE-DUAL PCI [5], AUGUSTUS [6], and ENTRUST-AF PCI [7], excluding the arm of the PIONEER AF-PCI [4] in which patients were treated with very low-dose rivaroxaban (2.5 mg twice daily), included 10,057 patients and showed that the use of the more potent P2Y12 inhibitors, prasugrel and ticagrelor, was associated with a significantly increased risk of major or clinically relevant bleeding compared to clopidogrel (RR 1.30; 95% CI 1.06–1.59; *p* = 0.01) [12], regardless of whether they were used in TAT or DAT. Importantly, there was no significant difference in the risk of major adverse cardiovascular events (MACE) between prasugrel or ticagrelor and clopidogrel (RR 1.02; 95% CI 0.57–1.82) [12].

Similar findings were reported in another meta-analysis comprising 22,014 patients from both prospective and retrospective studies that compared TAT and DAT, including the three randomized clinical trials, PIONEER AF-PCI [4], RE-DUAL PCI [5], and AUGUSTUS [6], which showed that, when analyzed collectively (i.e., both TAT and DAT regimens), ticagrelor (830 patients, 8%) and prasugrel (191 patients, 2%) were associated with a significantly higher incidence of bleeding compared to clopidogrel (9708 patients, 90%) (RR 1.36; 95% CI 1.18–1.57, and RR 2.11; 95% CI 1.34–3.30, respectively) [13]. Again, there were no significant differences in MACE incidence (RR 1.03; 95% CI 0.65–1.62 for ticagrelor and RR 1.49; 95% CI 0.69–3.24 for prasugrel) [13].

In the RE-DUAL PCI trial [2], which enrolled the highest proportion of patients treated with ticagrelor (327 patients, 12%), both in the TAT and DAT arms, a subgroup analysis was carried out based on the P2Y12 inhibitor used [14]. The benefit of DAT with dabigatran, at both 110 mg and 150 mg twice daily, in reducing the incidence of major or clinically relevant non-major bleeding events compared to conventional TAT with warfarin, aspirin, and P2Y12 inhibitor was confirmed, independent of whether ticagrelor or clopidogrel was used (Hazard Ratio [HR]; 95% CI 0.46; 0.28–0.76 vs. 0.51; 0.41–0.64, *p* of interaction 0.69 for dabigatran 110 mg twice daily, and 0.59; 0.34–1.04 vs. 0.73; 0.58–0.91, *p* of interaction 0.52 for dabigatran 150 mg twice daily) [14]. Similarly, ischemic events were not influenced by the choice between ticagrelor and clopidogrel [14].

In contrast to these findings, a significantly higher efficacy of prasugrel or ticagrelor was reported in a recent nationwide cohort study in Denmark [15]. Based on administrative data, 2259 AF patients on OAC undergoing PCI and claiming a prescription for P2Y12 inhibitor after discharge were identified. Of these patients, 1918 (85%) claimed a prescription for clopidogrel, 303 (13%) for ticagrelor, and 38 (2%) for prasugrel, presumably in a DAT setting [15]. Patients treated with ticagrelor or prasugrel were combined into one group comprising 341 individuals [15]. Whereas the 1-year bleeding risk between the two groups was comparable (RR 1.07; 95% CI 0.73–1.41; *p* = 0.69), that of MACE was significantly lower in the ticagrelor/prasugrel group compared with the clopidogrel group (RR 0.84; 95% CI 0.70–0.98; *p* = 0.02) [15].

In another study carried out in Japan, 5777 patients requiring OAC and undergoing PCI were retrospectively identified through an administrative database and a comparison of TAT regimens including either DOAC or vitamin K antagonist (VKA), aspirin, and clopidogrel or low-dose prasugrel (5 mg/day) was carried out [16]. Patients were divided into four subgroups: clopidogrel and DOAC (1628, 28%), clopidogrel and VKA (1334, 23%), low-dose-prasugrel and DOAC (1607, 28%), and low-dose prasugrel and VKA (1208, 21%) [16]. There was no significant difference in the incidence of death and gastrointestinal bleeding among the four subgroups [16]. The combination of low-dose prasugrel and DOAC was associated with lower incidence of myocardial infarction (OR 0.566, 95% CI 0.348–0.921), ischemic stroke (OR 0.701, 95% CI 0.502–0.979), and need for blood transfusions (OR 0.729, 95% CI 0.598–0.890) [16].

## 3. May Aspirin Be Used in DAT Instead of Clopidogrel?

While combinations of DOAC and various P2Y12 inhibitors are possible within a DAT regimen, current guidelines recommend clopidogrel as the antiplatelet agent of choice [1,2,3], based on the fact that clopidogrel was almost exclusively used in randomized clinical trials [4,5,6,7]. Aspirin was not included, although it was mentioned in previous ESC guidelines as a secondary option [17].

Clopidogrel and aspirin have different mechanisms and cannot be directly compared in terms of pharmacodynamics: clopidogrel irreversibly inhibits the P2Y12 receptor via hepatic activation and blocks ADP-mediated platelet aggregation, while aspirin irreversibly inhibits cyclo-oxygenase-1 (COX-1) and reduces thromboxane A2 synthesis [18] (Figure 2). Clopidogrel has long been considered more effective and comparably safe compared to aspirin [19]. In the randomized, international CAPRIE study, 19,185 patients with established atherosclerotic vascular disease, manifested as either recent ischemic stroke, recent myocardial infarction, or symptomatic peripheral arterial disease, were enrolled and assigned clopidogrel 75 mg once daily or aspirin 325 mg once daily [19]. Over a mean follow-up of 1.91 years, the cumulative risk of ischemic stroke, myocardial infarction, or vascular death was significantly lower in the group treated with clopidogrel (RR reduction 8.7%, 95% CI 0.3–16.5, *p* = 0.043) [19]. It is of note that the overall result was driven by a larger and significant effect only in the subgroup with symptomatic peripheral artery disease, so the true benefit may not be identical across the different subgroups as shown by the heterogeneity test (*p* = 0.042) [19]. No differences between the clopidogrel and aspirin groups were reported with regard to safety, particularly on bleeding [19]. On the basis of the results of the CAPRIE study [19], further and larger research has been carried out on the use of clopidogrel instead of aspirin in patients with established atherosclerotic cardiovascular disease, which confirmed that clopidogrel is at least as effective as aspirin, with similar bleeding risk. In particular, in a recent meta-analysis of five randomized control trials of clopidogrel versus aspirin as monotherapy in patients with established cardiovascular disease, where 26,855 patients were included, no statistically significant difference was observed between clopidogrel and aspirin in terms of all-cause mortality (OR 1.01, 95% CI 0.91–1.13, *p* = 0.83), ischemic stroke (OR 0.87, 95% CI 0.71–1.06; *p* = 0.16), and major bleeding rates (OR 0.77, 95% CI 0.56–1.06; *p* = 0.11). Patients receiving clopidogrel had borderline lower risk for MACE (OR 0.84, 95% CI 0.71–1.00; *p* = 0.05) and lower risk for non-fatal myocardial infarction (OR 0.83, 95% CI 0.71–0.97; *p* = 0.02) compared with patients receiving aspirin [20]. No differences were observed in terms of major bleeding [20]. In another recent experience, a patient-level meta-analysis of seven randomized trials comparing P2Y12 inhibitor monotherapy vs. aspirin in patients with CAD was carried out [21]. Out of the 24,325 patients included, 12,178 patients received P2Y12 inhibitor monotherapy, clopidogrel in the majority (62%), and 12,147 received aspirin [21]. Over 2 years, the risk of the primary outcome of combined cardiovascular death, myocardial infarction, and stroke, was significantly lower with P2Y12 inhibitor monotherapy compared with aspirin (HR 0.88; 95% CI 0.79–0.97; *p* = 0.012), mainly owing to less myocardial infarction (HR 0.77; 95% CI 0.66–0.90; *p* < 0.001) [21]. Major bleeding was similar (HR 0.87; 95% CI: 0.70–1.09; *p* = 0.23) and net adverse clinical events were lower (HR 0.89; 95% CI: 0.81–0.98; *p* = 0.020) [21]. The treatment effect was consistent across prespecified subgroups and types of P2Y12 inhibitors. Based on the above and other evidence, clopidogrel is now recommended in most recent guidelines as an alternative to aspirin in patients with chronic coronary syndrome [3].

The issue of clopidogrel non-responsiveness, and its association with an increased risk of adverse cardiovascular events in clinical practice, is well established [22,23,24]. It is known to affect up to 30% of patients and is associated with genetic, pharmacologic, or clinical factors (e.g., diabetes, renal failure, obesity) [22,23,24]. Also, clopidogrel may interact with atorvastatin, which is extensively used in patients with coronary artery disease, through a competition with the common metabolizer cytochrome P450 (CYP) 3A4 [25]. As a result, the efficacy of clopidogrel may be reduced and, therefore, the risk of cardiovascular events is increased [25]. This may be of relevance early after stent implantation, especially in patients who are not on DAPT but only on clopidogrel monotherapy in addition to DOAC for AF. Alternatives to clopidogrel may include ticagrelor, prasugrel (at either standard or low dose, although only very little data is available) as discussed above, and aspirin. While non-responsiveness to aspirin has also been reported in the literature, its clinical implications seem lower than for clopidogrel [26].

Not much evidence, however, is currently available for AF patients who, after the initial period of TAT post-PCI, are on a DAT regimen where aspirin is given instead of clopidogrel. In a search of the Korea National Health Insurance Service database from 2013 to 2020, a total of 9157 patients with AF who received DAT consisting of a DOAC and SAPT were identified [27]. Patients were then classified into the clopidogrel or aspirin group and 1:1 propensity score matching was performed, thereby identifying two groups consisting of 2882 patients each [27]. During a median follow-up of 20.1 months, the incidence of MACE was not significantly different between the clopidogrel and aspirin groups (HR 0.91, 95% CI 0.81–1.02) [27]. Neither was there a difference between groups in the incidence of major bleeding events (HR 0.94, 95% CI 0.78–1.12) and net adverse clinical events (HR 0.93, 95% CI 0.84–1.03) [27].

## 4. Discussion

Attempting to summarize the current evidence as reported above, it can be concluded that clopidogrel is the P2Y12 inhibitor of choice to be used in TAT. The higher risk of bleeding reported in several scenarios when prasugrel and ticagrelor were used makes their choice not recommended, bearing in mind that AF patients on OAC who undergo PCI are at high risk of bleeding [1,2,3,10]. Bleeding, in turn, even when not clinically relevant, has long been associated with high mortality after PCI, which is, in part, attributable to the partial or complete discontinuation of the ongoing antithrombotic therapy [28]. Whether the preference of clopidogrel over prasugrel and ticagrelor should be confirmed in DAT remains uncertain. No large and exclusive data focusing on this combination is available. The comparable benefit for both bleeding and thrombotic events reported in the RE-DUAL PCI sub-analysis [14] with DAT versus TAT, irrespective of the P2Y12 inhibitor used, clopidogrel or ticagrelor, is in support of the possible use in DAT of ticagrelor, which may be considered the most for patients at high thrombotic risk. While much less data is available for prasugrel, indirect data and pharmacological rationale may also make this choice an (alternative) option. The superior efficacy of prasugrel and ticagrelor, compared to clopidogrel when given in DAPT in patients with ACS, reported in the TRITON-TIMI 38 [8] and PLATO [9] studies, respectively, may justify preferring these drugs in patients with a history of a previous stent thrombosis or a previous coronary event during DAPT, or in stenting a potentially risky site of thrombosis (e.g., left main or last remaining vessel). An issue in choosing DAT with prasugrel or ticagrelor is the duration of the preceding TAT. Even though a short TAT with ticagrelor has been proposed in a recent consensus document from the European Heart Rhythm Association (EHRA) [29] in patients at very high thrombotic risk, the conventional combination of clopidogrel with DOAC and aspirin should be the standard. Because of the lack of data, and the possible risk of stent thrombosis and bleeding associated with this maneuver, it seems unjustifiable, at present, to switch from clopidogrel in TAT to prasugrel or ticagrelor in DAT.

With regard to aspirin, it may represent an alternative to clopidogrel in the DAT regimen. While additional evidence is awaited, the issue is about which patient should be suitable for continuing aspirin instead of clopidogrel after the initial TAT period. No data is available to guide such a decision, but, in addition to patients not responsive to clopidogrel at pharmacological or genetic testing [24], those who have experienced a previous coronary event or stent thrombosis while on clopidogrel, either in SAPT or DAPT, are likely good candidates. While the use of ticagrelor may be an alternative in the above scenarios, the lack of evidence in association with DOAC over several months make it less preferable to aspirin. Also, aspirin may be considered for patients in whom atorvastatin appears to be the only statin therapy that can be taken.

## 5. Conclusions

In patients on OAC, generally for AF, undergoing PCI, the initial DAPT regimen within TAT should generally include aspirin and clopidogrel. The use of more potent P2Y12 inhibitors, such as prasugrel or ticagrelor, is not recommended; however, they may likely be considered in patients who have confirmed or suspected clopidogrel non-responsiveness or have experienced previous coronary events while on clopidogrel, either in SAPT or DAPT. While the combination of DOAC and ticagrelor (or possibly prasugrel) may be reasonable during the early period after TAT has been concluded, the prolongation of this regimen is not supported by evidence and, therefore, should not be routinely adopted. Instead, based on indirect PCI data, pharmacological rationale, and emerging clinical experience, aspirin may be considered a valuable option as the antiplatelet component in DAT combination when clopidogrel non-responsiveness is confirmed or suspected and the occurrence of stent thrombosis may lead to catastrophic consequences. A possible algorithm incorporating all the considerations above is outlined in Figure 3.

## 6. Future Directions

Over the last years, progressive and extensive evidence has been produced regarding the issue of antithrombotic therapy in patients with AF on OAC undergoing PCI and, therefore, requiring DAPT. Several randomized trials have now firmly established that the current management should include an initial short course (up to one week) of TAT with DOAC, aspirin, and clopidogrel, followed by a period of 6–12 months of DAT with DOAC and clopidogrel, and then indefinite DOAC monotherapy [1,2,3]. The potential role of the more-potent P2Y12 inhibitors, like prasugrel and ticagrelor, should be investigated, at least, for having an opportunity of finding the, likely minimal, percentage of patients who cannot take clopidogrel for established non-responsiveness or previous cardiac events while on treatment. Whether the more potent P2Y12 inhibitors can be used in DAT beginning at the very initial moment after PCI—thus eliminating the current need for early TAT—would also be an interesting area of research. An even more adventurous line of research is the one investigating the temporary omission of OAC immediately after PCI and the administration of antiplatelet agents, either DAPT or even SAPT, only [30,31]. While OAC in AF patients is intended to mitigate the long-term risk of stroke and/or arterial embolism, temporary interruption is nonetheless contemplated in specific situations, such as when bleeding occurs or an elective invasive or surgical procedure is programmed [29,32]. In the MATRIX-2 trial [30], 3010 AF patients who have undergone successful PCI are being randomized to P2Y12 inhibitor monotherapy (clopidogrel, ticagrelor, or prasugrel) or standard of care therapy (i.e., initial and short TAT followed by 6–12 months of DAT) for the first month post-procedure, to be then switched to DOAC monotherapy for the following 11 months. In the WOEST-3 trial [31], 2000 AF patients undergoing PCI are being randomized to guideline-directed therapy (edoxaban plus P2Y12 inhibitor plus limited duration of aspirin) or a 30-day DAPT strategy (P2Y12 inhibitor plus aspirin) followed by DAT (edoxaban plus P2Y12 inhibitor) for the subsequent 5–11 months. In both arms, selection of the P2Y12 inhibitor is at the discretion of the treatment physician [31].

## Figures and Tables

**Figure 1 jcm-14-06331-f001:**
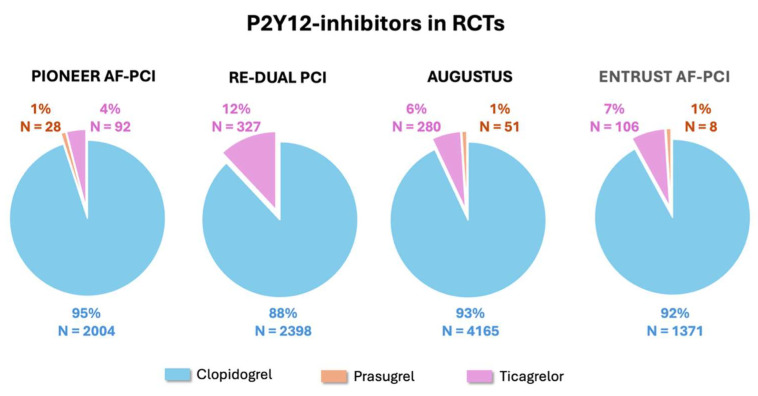
Proportions of the use of the various P2Y12 inhibitors in the four randomized clinical trials comparing TAT vs. DAT in patients with AF undergoing PCI and/or hospitalized for ACS [4,5,6,7]. ACS: acute coronary syndrome; AF: atrial fibrillation; DAT: double antithrombotic therapy; PCI: percutaneous coronary intervention; TAT: triple antithrombotic therapy.

**Figure 2 jcm-14-06331-f002:**
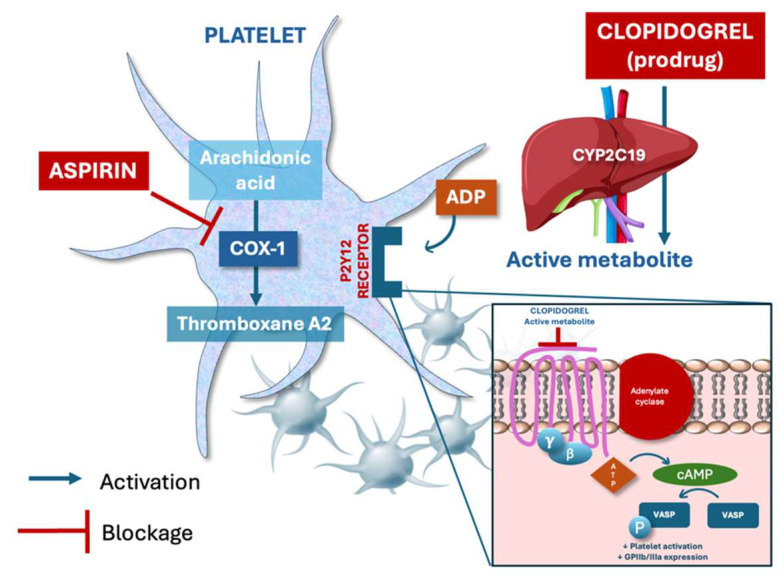
Mechanisms of action of aspirin and clopidogrel on platelet inhibition. Aspirin irreversibly inhibits cyclooxygenase-1 (COX-1), blocking the synthesis of thromboxane A_2_, a potent promoter of platelet aggregation and vasoconstriction. Clopidogrel (prodrug) is converted in the liver to an active metabolite that irreversibly binds to the P2Y_12_ receptor, inhibiting the ADP-mediated signaling pathway, and thus reducing GPIIb/IIIa expression and platelet aggregation. VASP: Vasodilator-Stimulated Phosphoprotein.

**Figure 3 jcm-14-06331-f003:**
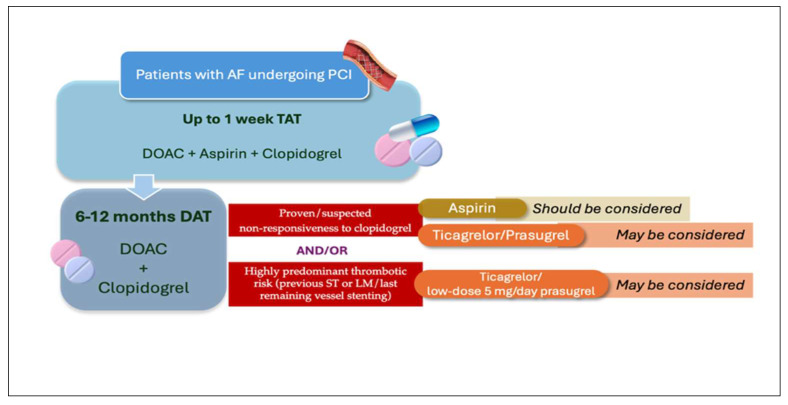
Algorithm including different possibilities of DAT regimen after the first week of TAT in patients with AF undergoing PCI. AF: atrial fibrillation; DAT: double antithrombotic therapy; DOAC: direct oral anticoagulant; LM: left main; PCI: percutaneous coronary intervention; ST: stent thrombosis; TAT, triple antithrombotic therapy.

**Table 1 jcm-14-06331-t001:** Design of the four randomized clinical trials TAT vs. DAT in patients with AF undergoing PCI and/or hospitalized for ACS [4,5,6,7]. ACS: acute coronary syndrome; AF: atrial fibrillation; BMS: bare metal stent; CRNM: clinically relevant non-major bleeding; DAT: double antithrombotic therapy, DAPT: dual antiplatelet therapy; DOAC: direct oral anticoagulant; DES: drug eluting stent; ISTH: International Society on Thrombosis and Hemostasis; OD: once daily; BID: bis in die; PCI: percutaneous coronary intervention; P2Y12i: P2Y12 inhibitor; TAT: triple antithrombotic therapy; TIMI: Thrombolysis In Myocardial Infarction; VKA: vitamin K antagonist.

Randomized Clinical Trial	PIONEER	RE-DUAL	AUGUSTUS	ENTRUST-PCI
**DOAC**	Rivaroxaban	Dabigatran	Apixaban	Edoxaban
**Inclusion criteria**	AF + PCI (including stent implantation)	AF + PCI	AF + ACS and/or PCI	AF + PCI (including stent implantation)
**Duration**	12 months	Minimum 6 months	6 months	12 months
**DOUBLE arm**	Rivaroxaban 15 mg OD + P2Y12i	Dabigatran 150 mg BID + P2Y12i Dabigatran 110 mg BID + P2Y12i	Apixaban 5 mg BID + P2Y12i VKA + P2Y12i	Edoxaban 60 mg OD + P2Y12i
**TRIPLE arm**	Rivaroxaban 2.5 mg BID + DAPT VKA + DAPT	VKA + DAPT	Apixaban 5 mg BID + DAPT VKA + DAPT	VKA + DAPT
**ASA duration in DAPT arm**	1, 6 or 12 months	1 month for BMS, 3 months for DES	6 months	Risk-based: 30 days to 12 months
**Primary endpoint**	TIMI clinically significant bleeding or bleeding requiring medical attention	Time to first ISTH major or CRNM bleeding event	Time to first ISTH major or CRNM bleeding at 6 months	Time to first ISTH major or CRNM bleeding event

## Data Availability

No new data were created or analyzed in this study.

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
