# Peer review of "Antiplatelet Therapy in Atrial Fibrillation Patients on Direct Oral Anticoagulants Undergoing Percutaneous Coronary Intervention: Which and How"

_jcm, 2025, doi:10.3390/jcm14176331_

Round 1

Reviewer 1 Report

Comments and Suggestions for Authors

In the current manuscript, the authors assess optimal antiplatelet therapy in patients with atrial fibrillation (AF) on oral anticoagulation (OAC) undergoing percutaneous coronary intervention (PCI). The paper reviewed the evidence from randomized controlled trials (RCTs), meta-analyses, and registries. It explored the role of aspirin and P2Y12 inhibitors in different phases of therapy (triple vs. double antithrombotic therapy). The clinical guidelines are changing rapidly about this topic, and there is a significant interest in individualized antithrombotic strategies. Including a simplified algorithm improves its usefulness in clinical practice. Is the review a narrative synthesis rather than a systematic review? There is no detailed description of the search strategy, inclusion/exclusion criteria, or how studies were selected and weighted. If systematic elements were applied, provide PRISMA-style details. Please include clear and detailed comments on the quality and certainty of the evidence. Please tone down your recommendation regarding the DAPT, as it is not strongly supported by evidence. Please also include a table summarizing bleeding vs. ischemic risks for potent P2Y12 inhibitors in the AF+PCI population. Please include evidence-based and the authors’ proposed recommendations and grades in Figure 3 (you may change your figure with color-coded boxes according to the evidence). In the “Future Directions” section, the author’s proposal about omitting OAC entirely post-PCI lacks adequate discussion for potential stroke risk. Please revise your title about the patient group - AF pts undergoing PCI (also, there was no data about the VKA, thus please change the title to novel OACs or DOACs?). The comparison of aspirin with clopidogrel in dual therapy is speculative rather than powerful evidence-based data.  

Author Response

RESPONSE TO REVIEWER 1

In the current manuscript, the authors assess optimal antiplatelet therapy in patients with atrial fibrillation (AF) on oral anticoagulation (OAC) undergoing percutaneous coronary intervention (PCI). The paper reviewed the evidence from randomized controlled trials (RCTs), meta-analyses, and registries. It explored the role of aspirin and P2Y12 inhibitors in different phases of therapy (triple vs. double antithrombotic therapy). The clinical guidelines are changing rapidly about this topic, and there is a significant interest in individualized antithrombotic strategies. Including a simplified algorithm improves its usefulness in clinical practice.

COMMENT 1: Is the review a narrative synthesis rather than a systematic review? There is no detailed description of the search strategy, inclusion/exclusion criteria, or how studies were selected and weighted. If systematic elements were applied, provide PRISMA-style details. Please include clear and detailed comments on the quality and certainty of the evidence.

RESPONSE 1: We thank the reviewer for pointing this out. Indeed, ours is a narrative review where no formal search strategy has been applied while various combinations of the search terms including “atrial fibrillation”, “percutaneous coronary intervention”, “acute coronary syndrome”, “chronic coronary syndrome”, “antiplatelet”, “anticoagulant”, and “antithrombotic therapy” where used instead. In addition, the paper was not intended to be a consensus paper which might have warranted some grading of the evidence. In this case however, it would not have been appropriate to use the levels commonly used in Guidelines, that is A for data derived form multiple randomized clinical trials or meta-analyses, B for data derived from a single randomized clinical trial or large non-randomized studies, and C for consensus of opinion of the experts, retrospective studies, registries. Even in consensus documents from Associations and Working Groups, grading of evidence is not provided as above, but rather defined in an individual fashion (see for example Atar D et al. Management of patients with congenital bleeding disorders and cardiac indications for antithrombotic therapy. Eur Heart J Cardiovasc Pharmacother. 2025 May 2;11(3):275-289. doi: 10.1093/ehjcvp/pvaf006. PMID: 40145128 where the evidence was graded in a scale of 4 as 1/4, 2/4, 3/4 and 4/4). Again, being our paper a review and not a consensus paper, we thought not appropriate to give level of evidence of our statements with the strength of the studies being derived instead from the description of their design and size the text.

COMMENT 2: Please tone down your recommendation regarding the DAPT, as it is not strongly supported by evidence.

RESPONSE 2: While thanking the reviewer for this comment, we can’t exactly see what/where in the text he/she is referring about. We would be glad to work on that should we have more information. We would like however to remark that we do not formulate new recommendations compared to current Guidelines as regards established therapies in patients with AF undergoing PCI (see for example Van Gelder IC et al. 2024 ESC Guidelines for the management of atrial fibrillation developed in collaboration with the European Association for Cardio-Thoracic Surgery (EACTS).Eur Heart J. 2024 Sep 29;45(36):3314-3414. doi: 10.1093/eurheartj/ehae176. PMID: 39210723), including indication and duration of DAPT. We only focus on and discuss the issue of the choice of the different antiplatelet agents, especially for less investigated and therefore cumbersome scenarios, such as, the presence of ascertained or suspected non-responsiveness to clopidogrel.

COMMENT 3: Please also include a table summarizing bleeding vs. ischemic risks for potent P2Y12 inhibitors in the AF + PCI population.

RESPONSE 3: We thank the reviewer for this relevant comment. Unfortunately however, we are unable to address it since no dedicated data are reported in the pivotal randomized clinical trials for the (small) populations receiving ticagrelor/prasugrel instead of clopidogrel. In the meta-analysis by Casula M et al. Meta-analysis comparing potent oral P2Y12 inhibitors versus clopidogrel in patients with atrial fibrillation undergoing percutaneous coronary intervention. Am J Cardiovasc Drugs. 2021 Mar;21(2):231-240. doi: 10.1007/s40256-020-00436-8. PMID: 32895853) the incidence of both bleeding and ischemic events can be derived and the relative risks of bleeding and ischemic events are indeed reported in the text. Building also a table for this, appears to us redundant and therefore not of very much use for the reader.

COMMENT 4: Please include evidence-based and the authors’ proposed recommendations and grades in Figure 3 (you may change your figure with color-coded boxes according to the evidence).

RESPONSE 4: We thank again the reviewer for rising this highly relevant point. However as pointed above, ours is a narrative review focusing on the only aspect of choosing the antiplatelet agent to combined in triple or double therapy and dealing with very limited evidence. Our intention was therefore to deliberately not grade the level of evidence. Ours is not a consensus paper endorsed but only a review and discussion of a topic which  nowadays may be of importance given the choices we have in selecting different antiplatelet agents. By the way, we chose to adopt statements “should be” and “may be” that in Guidelines are associated to the classes of recommendations “IIa" and “IIb” respectively. Experts may find inappropriate to use the terminology above given that they are generally reserved to Guidelines, but we thought that this may be a way to make our considerations and suggestions more understandable and useful.

COMMENT 5: In the “Future Directions” section, the author’s proposal about omitting OAC entirely post-PCI lacks adequate discussion for potential stroke risk.

RESPONSE 5: We thank the reviewer for the comment. We would like however to remark that temporary omission of OAC is not our proposal, but is instead a current line of research with two important trials, namely MATRIX-2 (Windecker S, Valgimigli M. Monotherapy With P2Y12 Inhibitors in Patients With Atrial fIbrillation Undergoing Supraflex Stent Implantation (MATRIX-2). https://clinicaltrials.gov/study/NCT05955365 (Accessed July 5, 2025) and WOEST-3 (Verburg A, et al. Temporary omission of oral anticoagulation in atrial fibrillation patients undergoing percutaneous coronary intervention: rationale and design of the WOEST-3 randomised trial. EuroIntervention. 2024 Jul 15;20(14):e898-e904. doi: 10.4244/EIJ-D-24-00100. PMID: 39007830) being underway. We added nonetheless in the text of the “Future Directions” section a short consideration on the potential risk of stroke and/or arterial thromboembolism when (temporarily) omitting OAC in these patients.

COMMENT 6: Please revise your title about the patient group - AF pts undergoing PCI (also, there was no data about the VKA, thus please change the title to novel OACs or DOACs?).

RESPONSE 6: We thank the reviewer for pointing this out. We have edited the title in accordance, in hope that this new version is acceptable also for the guest editor of the special issue and the editorial office as well who were there ones suggesting the initial title.

COMMENT 7: The comparison of aspirin with clopidogrel in dual therapy is speculative rather than powerful evidence-based data.  

RESPONSE 7: While thanking the reviewer for the comment, we indeed agree that no many data are available with aspirin vs. clopidogrel in double therapy. Accordingly we put a question mark at the end of the title of section discussing this issue (see, May Aspirin be Used in DAT instead of Clopidogrel?). This combination is at present speculative but a first piece of data has been recently published (Kim SH et al.. Clinical outcomes with the use of aspirin versus clopidogrel as a combination therapy with direct oral anticoagulant after coronary stent implantation in patients with atrial fibrillation Eur Heart J Cardiovasc Pharmacother 2025 May 29:pvaf043. doi: 10.1093/ehjcvp/pvaf043. Online ahead of print) and we are confidentially aware of other similar initiative being planned. We believe therefore that this is Smay even further be in the future, an issue of interest.

Reviewer 2 Report

Comments and Suggestions for Authors

This is a well written review by the authors on an important topic.  The manuscript is quite comprehensive and I have no issues with the data and interpretation by the authors.  Following I have one comment.

Comment:

It is worth discussing that in the AF population oral anticoagulation is to address the long term cumulative risk of stroke/arterial thromboembolism, however can be temporarily paused for bleeding or if bleeding risk is too high on anti platelet therapy after PCI.  I would mention since AF patients also have related procedures that require mandatory uninterrupted oral anticoagulation such as DCCV (1 month) or catheter ablation (2-3 months).  Please cite expert consensus document on management of AF and catheter ablation (PMID: 38609733).

Author Response

RESPONSE TO REVIEWER 2

This is a well written review by the authors on an important topic. The manuscript is quite comprehensive and I have no issues with the data and interpretation by the authors.  Following I have one comment.

We thank very much h the reviewer for his/her appreciation of our work.

COMMENT 1: It is worth discussing that in the AF population oral anticoagulation is to address the long term cumulative risk of stroke/arterial thromboembolism, however can be temporarily paused for bleeding or if bleeding risk is too high on anti platelet therapy after PCI.  I would mention since AF patients also have related procedures that require mandatory uninterrupted oral anticoagulation such as DCCV (1 month) or catheter ablation (2-3 months). Please cite expert consensus document on management of AF and catheter ablation (PMID: 38609733).

RESPONSE 1: We thank the reviewer for the comment. We added a sentence in the “Future Directions” section mentioning the issue raised and referencing it (Steffel J et al. 2021 European Heart Rhythm Association Practical Guide on the Use of Non-Vitamin K Antagonist Oral Anticoagulants in Patients with Atrial Fibrillation.Europace. 2021 Oct 9;23(10):1612-1676. doi: 10.1093/europace/euab065. PMID: 33895845; and Tzeis S et al. 2024 European Heart Rhythm Association/Heart Rhythm Society/Asia Pacific Heart Rhythm Society/Latin American Heart Rhythm Society expert consensus statement on catheter and surgical ablation of atrial fibrillation. J Arrhythm. 2024 Oct 6;40(6):1217-1354. doi: 10.1002/joa3.13082). We have decided however not to include  the scenario of procedures requiring uninterrupted oral anticoagulation for some time afterwards (such 1 month after cardioversion or 2-3 months after catheter ablation) since it appeared to us not fully pertinent to article topic. As suggested however, we have cited the consensus paper by Tzeis S et al. 2024 European Heart Rhythm Association/Heart Rhythm Society/Asia Pacific Heart Rhythm Society/Latin American Heart Rhythm Society expert consensus statement on catheter and surgical ablation of atrial fibrillation.J Arrhythm. 2024 Oct 6;40(6):1217-1354. doi: 10.1002/joa3.13082. eCollection 2024 Dec. PMID: 39669937). 

Round 2

Reviewer 1 Report

Comments and Suggestions for Authors

The authors reasonably and partially replied to my previous queries. I have no further comments.